# Adjustment of the Grass Fuel Moisture Code for Grasslands in Southern Brazil

**João Francisco Labres dos Santos \***  **, Bruna Kovalsyki, Tiago de Souza Ferreira, Antonio Carlos Batista and Alexandre França Tetto**

Forest Engineering Department, Agrarian Sector, Paraná Federal University, Av. Prefeito Lothário Meissner, 632, Curitiba 80210-170, PR, Brazil
* Correspondence: joaolabres@ufpr.br

**Abstract:** Grasslands are one of the vegetation types most widely affected by wildfires in southern Brazil. It is a fire-dependent ecosystem and it is necessary to know the hourly fuel moisture variation for its management. The objective of this work is to fit Grass Fuel Moisture Code (GFMC) models to estimate the moisture content for the grassland of the State Park of Vila Velha, Paraná, Brazil. Data sampling to determine the hourly moisture content was performed during the winter of 2018 and divided into two campaigns of five days with stable weather conditions. Destructive samples were taken out for the sorption tests on climatic chambers to obtain the equilibrium moisture content and the time lag values. The fitted equilibrium moisture and time lag models were evaluated by residual distribution analysis, mean absolute error (MAE), root mean square error (RSME) and coefficient of determination ($R^2$). The fitted model performed better than the original GFMC model due to the obtained MAE, RSME and $R^2$ values. The results showed that the fitted GFMC model is better to predict the fine fuel moisture for the region.

**Keywords:** wildfires; moisture content; time lag; equilibrium moisture content

## 1. Introduction

Grass fuel belongs to a class of important dangerous fuels due to its continuity and complete exposure to the sun [1,2]. In addition to these characteristics, standing dead grass dries more quickly than other fuels and its moisture content responds instantaneously to changes in weather [1,2].

The Brazilian south grassland occurs naturally from the State of Rio Grande do Sul to Santa Catarina, and to a smaller extent in the State of Paraná [3]. It can be divided into two categories: the Pampa grassland which occurs in southern Rio Grande do Sul, and the highland grassland, located on the South Brazilian highland plateau in northern Rio Grande do Sul, Santa Catarina and Paraná [4].

The highland grasslands are a fire-dependent ecosystem used for the conservation of its species and landscape; they have been traditionally managed with fire for its renovation [4]. Therefore, due to the presence of fire in this ecosystem, it is necessary to use accurate information about the fine fuel moisture content.

Fuel moisture content is the amount of water in vegetable biomass and it is a critical variable to estimate the fire ignition and propagation danger [5]. The process associated with the changes of dead fine fuel moisture is related to two variables: precipitation and the atmosphere's vapor pressure. In the absence of rainfall or dew condensation, the fuel moisture exchange in vapor states occurs via the isothermal processes of adsorption and desorption [6].

When dead fine fuel is introduced to an environment at a constant temperature and a relative humidity, the fuel moisture either increases or decreases until it reaches a constant value called the equilibrium moisture content (*EMC*). Byram [7] describes that

the adsorption and desorption processes occur at a log drying rate designated by time lag (τ). The time lag is the time required for a 63.2% increase or decrease in the remaining evaporable water in the forest fuel.

The Fine Fuel Moisture Code (FFMC) was originally developed by Van Wagner in 1974 as part of the Fire Weather Index to model the litter moisture in *Pinus bankisiana* and *Pinus ponderosa* stands in Canada using the weather meteorological data collected at noon [8]. The estimates provided by the FFMC are based on the concepts of equilibrium moisture content and time lag. However, dead fine fuel dries or wets quickly enough to use hourly weather measurements [8]. The Hourly Fine Fuel Moisture Code (HFFMC) is a method developed for computing fine fuel moisture content throughout the day using hourly weather information with a similar structure to the FFMC [9].

The HFFMC models estimate the moisture content in a closed canopy conifer forest stand, compensating for the effects caused by rain and solar radiation, as well as for the reduction in the surface wind caused by the forest stand [1]. Therefore, Wotton [1] introduced the effect of grass exposure to solar radiation using the concept of fuel temperature, obtaining good results in the grass fuel moisture estimates in Canada.

The moisture content modelling process can be separated into field sampling, laboratory testing and data processing [6]. It is an important factor in the structure of danger indices and constitutes a great database for studies in remote sensing [10].

White [5] pointed out that there are a lack of studies regarding fuel moisture content modelling in Brazil. Alves [11] carried out a modeling study into the moisture content in *Pinus elliotti* stands in the Paraná state; however, they disregarded the concepts of equilibrium moisture content and time lag. In this sense, the hypothesis tested was the adjustment of the Grass Fuel Moisture Code models, which could bring about better results to estimate the grass moisture content in order to subsidize fire danger research and fire management in Southern Brazil.

## 2. Materials and Methods

### 2.1. Study Site

This study was performed in a highland grassland area located in the Vila Velha State Park, Paraná, Brazil (25.24° S–50.00° W). The region's climate is classified as temperate (Cfb), according to the Köppen classification, with the average temperature, of the coldest month, below 18 °C, without a defined dry season. August is the driest month of the year with an average rainfall of 78 mm [12,13]. The sampling was carried out with a transition between the highland grassland and the Cerrado (Brazilian savanna) with the presence of grass species, such as *Paspalum notatum* Flüggé, *Axonopus fissifolius* (Raddi) Kuhlm, *Andropogon lateralis* Ness and *Stipa* spp. [3].

### 2.2. Field Sampling

Both the field sampling and the weather measurements were carried out using methodological procedures developed by Wotton [1]. An open site was chosen away from any standing trees, which could influence the weather conditions. Two five-day sampling campaigns were performed in the winter of 2018 to assemble the validation dataset of grass moisture. A standard weather station owned by the Sistema Meteorológico do Paraná [13] provided the weather measurements. A simple fuel inventory was performed to determine the fuel density before the hourly moisture sampling.

Destructive sampling was conducted throughout the day from 8 a.m. to 5 p.m. at 1h intervals. This period was chosen due to the limitations of the park's opening hours and the duration of daylight. Thus, three hermetic packages with approximately 50 g of wet mass were sampled at each interval. The samples were weighed with a digital scale at the field site and transported to the Forest Fires Laboratory of the Paraná Federal University.

### 2.3. Laboratory Tests

The samples obtained were dried in an oven at 75 °C until a constant mass was achieved. Three subsamples weighing 100 g were prepared for the sorption tests. The equilibrium moisture content (*EMC*) was determined in a climatic chamber controlled by a PIC TM4C1294 from Texas Instruments Incorporation with a potential air temperature of 10–40 °C and relative humidity of 25–95%. Next, two digital scales were built with a precision of 0.5 g on the Arduino free platform in C/C++ language, with a serial output and a port power on a Universal Serial Bus (USB) to register the sample mass variation. The chamber temperature was fixed at 26.7 °C and the relative humidity ranged from 30% to 90% with intervals of 10%, according to the methodological procedures developed by Van Wagner [14]. All moisture contents were determined by the gravimetrical method (moisture content by dry mass).

The sorption tests were performed at approximately 72 h at each RH interval or until no mass variation. The registration was carried with a periodicity of 5 min until the fuel mass reached the equilibrium. The mean of the last 4 h was considered as the equilibrium moisture content for each sample.

### 2.4. Modelling of Equilibrium Moisture Content

This stage was developed using the methodologies proposed by Lopes [6] and Bakšić [9]. The sorption process can be described through a pure exponential equation. Byram's [7] pure exponential model (1) and the one-term model (2) of Henderson and Pabis [6] were used to model the drying and wetting cycle:

$$E_{(t)} = e^{-\tau t} = \frac{m_t - EMC}{m_0 - EMC} \tag{1}$$

$$E_{(t)} = ae^{-\tau t} = a\left(\frac{m_t - EMC}{m_0 - EMC}\right) \tag{2}$$

In which *E* is the fraction of evaporable water remaining in the fuel sample, *t* is time (h), *e* is Euler's number, *m* is the moisture content (at the beginning of the drying process and at time *t*), *EMC* is the equilibrium moisture content, $\tau$ is the time lag and *a* is a coefficient. The time lag is the time that the fuel needs to lose or gain 63.2% of the evaporable moisture until reaching equilibrium. This value is assumed to be a constant in order to obtain simpler calculations for the desorption and adsorption process. A combination of several exponential curves occurs during the sorption process [15]. Then, the sorption curves obtained in this study were divided in four time lags, in which *E* is equal to 0.368, 0.135, 0.050 and 0.018 for the adsorption and 0.632, 0.865, 0.950 and 0.982 for desorption, respectively.

Next, an *EMC* model was proposed by Van Wagner, to estimate the equilibrium moisture content for several fuels, as follows:

$$EMC = \alpha H^\beta + \gamma e^{\frac{H-100}{\delta}} \tag{3}$$

In which *EMC* is the equilibrium moisture content, *H* is RH, *T* is temperature (°C), *e* is Euler's number, and $\alpha$, $\beta$, $\gamma$, $\delta$ are constants. Next, the mean absolute error (MAE), the root mean squared error (RMSE) and the coefficient of determination ($R^2$) were calculated to evaluate the fitting quality of an analysis of residual distributions plots.

### 2.5. Development of the Grass Fuel Moisture Code

The GFMC model had some peculiarities in its calculation. The temperature and relative humidity are corrected before the moisture estimation due to the exposure of grass fuel to solar radiation [1]. The grass code calculation is based on two stages: the increase in moisture by rainfall and the exponential drying and wetting process (Equations (1) and (2)). Only the second stage was considered in this work. The determination of the solar radiation influence on the temperature and the RH is better described in Wotton [1]. The moisture

content ($m_t$) in drying conditions is calculated from the value of the previous hours moisture content ($m_0$)). The same procedure is performed for wetting conditions. Next, the *EMC* is divided into two equations for the drying and wetting conditions to estimate the moisture content. Therefore, it is necessary to calculate log drying ($k_d$)) and wetting (($k_w$) rates based on the temperature, RH and wind speed (Equations (4) and (5)):

$$k_d = \alpha e^{-0.0365T}[0.424\left(1 - \left(\frac{H}{100}\right)^{1.7}\right) + 0.0694W^{0.5}\left(1 - \left(\frac{H}{100}\right)^8\right)] \tag{4}$$

$$k_w = \beta e^{-0.0365T}[0.424\left(1 - \left(\frac{100-H}{100}\right)^{1.7}\right) + 0.0694W^{0.5}\left(1 - \left(\frac{100-H}{100}\right)^8\right)] \tag{5}$$

In which $W$ is wind speed (km·h$^{-1}$), $e$ is Euler's number and $\alpha$ and $\beta$ are the coefficients for the drying and wetting rates, respectively. The full procedure for the GFMC model calculation is shown in Table 1.

**Table 1.** Time lag and standard deviation at a temperature of 26.7 °C.

| Process | Tests (n) | $\tau_1$ (h) | | $\tau_2$ (h) | | $\tau_3$ (h) | | $\tau_4$ (h) | |
|---|---|---|---|---|---|---|---|---|---|
| | | Mean | $S^2$ | Mean | $S^2$ | Mean | $S^2$ | Mean | $S^2$ |
| Desorption | 21 | 2.21 | 1.21 | 2.81 | 1.52 | 4.89 | 2.92 | 5.02 | 3.22 |
| Adsorption | 21 | 3.39 | 2.12 | 3.47 | 2.54 | 6.33 | 2.1 | 2.8 | 4.33 |

The modifications carried out on the GFMC models were the change in coefficients from the *EMC* models and the drying/wetting rate equations when considering only the first time lag of each process. The estimates from the new GFMC model were evaluated using the residual plot analysis, mean absolute error (MAE), the root mean squared error (RMSE), the mean bias error (MBE) and the Pearson's correlation tests [1,9].

## 3. Results

### 3.1. Sorption Tests

An exponential behavior was found for all the sorption tests with quick decreases and increases in the moisture content at the beginning and slowly reaching the *EMC* at the end of the sorption process. Table 1 contains the four time lag periods for each curve at a temperature of 26.7 °C with their respective means and standard deviations ($s^2$).

The curves of fitted models, as well their moisture content values for the desorption and adsorption process are plotted in Figure 1.

The fitted models for determining time lag and their coefficients are presented in Table 2.

**Table 2.** Exponential sorption models adjusted to a temperature of 26.7 °C and statistical parameters.

| Model | Desorption | | | | | Adsorption | | | | |
|---|---|---|---|---|---|---|---|---|---|---|
| | $\tau$ | $a_1$ | $R^2$ | MAE | RSME | $\tau$ | $a_1$ | $R^2$ | MAE | RSME |
| Byram (1963) | 0.31 | - | 0.828 | 0.092 | 0.126 | 0.217 | - | 0.862 | 0.082 | 0.117 |
| Henderson and Pabis (1961) | 0.33 | 1.065 | 0.83 | 0.09 | 0.125 | 0.241 | 1.103 | 0.869 | 0.082 | 0.114 |

Both models show a good fit with the data obtained in the sorption tests. The Byram's model show $R^2$ values of 0.8332 and 0.8676, while the one-term model show a $R^2$ value of 0.8318 and 0.8689 for desorption and adsorption, respectively. The MAE values obtained by Byram's model were 0.0919 for desorption and 0.0822 for adsorption. The RMSE for each process was 0.1254 for desorption, as well as 0.1169 for adsorption. MAE values of 0.0904 and 0.0816 for desorption and adsorption were found for the one-term model. The RMSE values obtained were 0.1245 for desorption and 0.1138 for adsorption. The residuals

for both models show a random distribution. Thus, the one-term model was chosen for this study due to presenting a lower RMSE value.

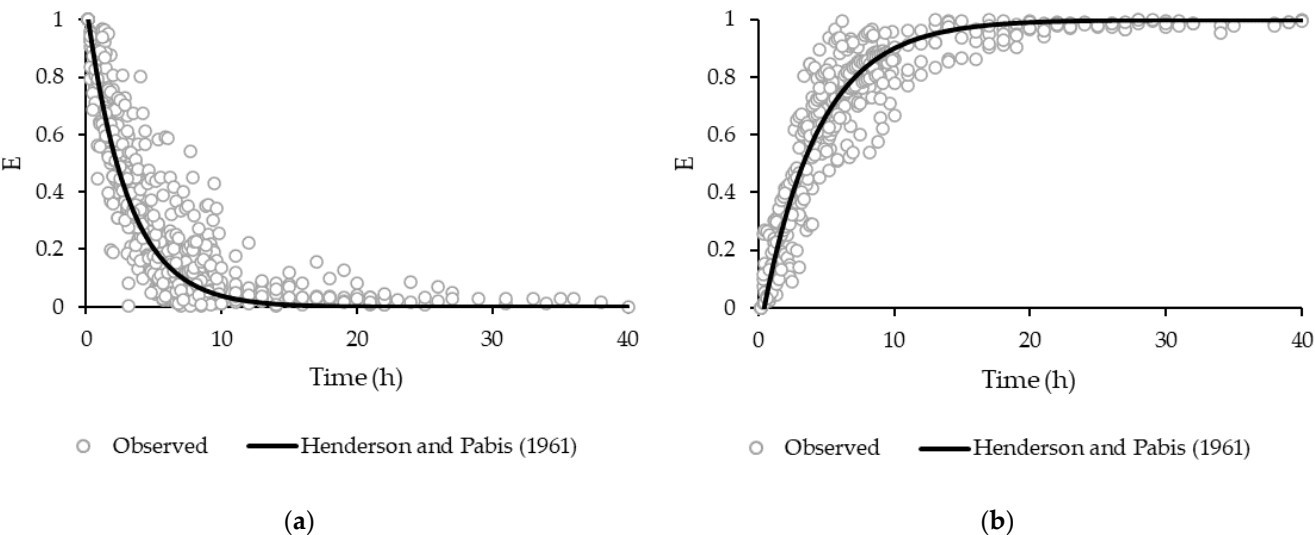

**Figure 1.** (**a**) Desorption and (**b**) adsorption curves obtained by fitted models.

Table 3 shows the coefficients estimated for the *EMC* models for sorption processes and its respective parameters of fit quality.

**Table 3.** Exponential sorption models adjusted to a temperature of 26.7 °C and statistical parameters.

| Process | Estimated Coefficients | | | | Evaluation Paramenters | | |
| --- | --- | --- | --- | --- | --- | --- | --- |
| | α | β | γ | δ | MAE | RMSE | R² |
| Desorption | 1.0034 | 0.6374 | 13.1007 | 23.1046 | 0.1632 | 0.2039 | 0.9993 |
| Adsorption | 0.7973 | 0.7039 | 9.3548 | 12.6231 | 0.3008 | 0.3696 | 0.9969 |

The Van Wagner's *EMC* model shows a good fitting ability for both the sorption data. The fitted model obtained $R^2$ values of 0.9993 and 0.9969 for desorption and adsorption data, respectively. In addition, MAE values of 0.1632 were found for desorption and 0.3008 for adsorption *EMC*. The RMSE obtained for desorption *EMC* was 0.2040 and 0.3696 for adsorption. It is noteworthy that the experiments for the effect of temperature were not carried out in this study. Thus, Van Wagner's (1987) approach was adopted by applying an additional function based on a temperature of 26.7 °C in the final model. Thus, the desorption and adsorption *EMC* models then took the following form with the obtained coefficients:

$$EMC_d = \left(1.0034H^{0.6374} + 13.1007e^{\frac{H-100}{23.1046}}\right) + 0.27(26.7 - T)\left(1 - e^{-0.115H}\right) \quad (6)$$

$$EMC_w = \left(0.7973H^{0.7039} + 9.3548e^{\frac{H-100}{12.2548}}\right) + 0.27(26.7 - T)\left(1 - e^{-0.115H}\right) \quad (7)$$

In which *EMC* is the equilibrium moisture content, *H* is RH, *T* is temperature (°C) and *e* is Euler's number. The fitted *EMC* curves plotted with the relative humidity are shown in Figure 2.

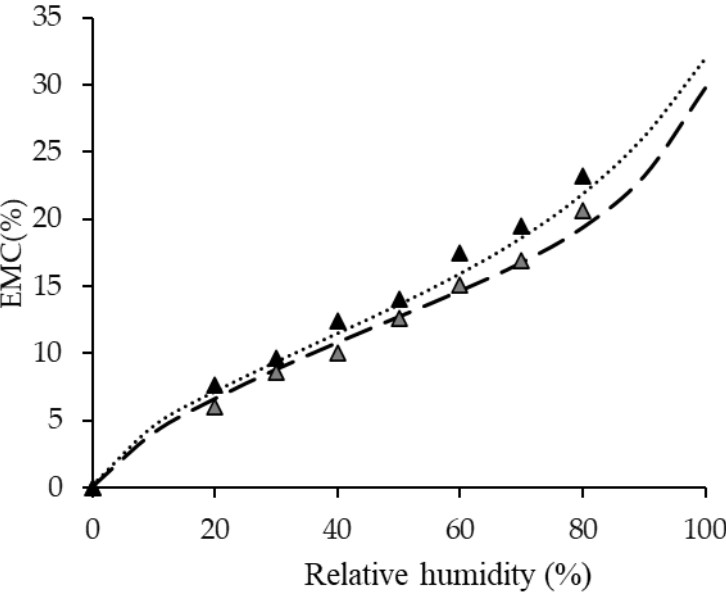

— — Adsorption est.   ········ Desorption est.

▲ Adsorption obs.   ▲ Desorption obs.

**Figure 2.** *EMC* data of adsorption and desorption for Vila Velha State Park grassland with the fitted curves.

It is possible to observe that the adsorption and desorption estimations show the expected behavior from a sigmoid curve. Therefore, the *EMC* is zero when the RH approaches zero. The presence of the hysteresis average of 1.32% between the adsorption and desorption curves is another factor observed.

The changes carried out in the response time equations led to the model taking the following form:

$$k_d = 0.3299e^{-0.0365T}[0.424\left(1 - \left(\frac{H}{100}\right)^{1.7}\right) + 0.0694W^{0.5}\left(1 - \left(\frac{H}{100}\right)^{8}\right)] \qquad (8)$$

$$k_w = 0.2415e^{-0.0365T}[0.424\left(1 - \left(\frac{100-H}{100}\right)^{1.7}\right) + 0.0694W^{0.5}\left(1 - \left(\frac{100-H}{100}\right)^{8}\right)] \quad (9)$$

In which $T$ is temperature (°C), $H$ is RH, $W$ is wind speed (km·h$^{-1}$), $k_d$ is the drying response rate (h$^{-1}$), $k_w$ is the wetting response rate (h$^{-1}$) and $e$ is Euler's number.

### 3.2. Sorption Tests

The moisture content obtained from field samples ranged from 5.6 to 146.7%. No rain precipitation occurred during the sampling period. However, values above extinction were observed (Figure 3).

The values estimated by the original and the fitted GFMC model are close to those obtained in the field. Through a linear regression analysis, it was possible to verify that the fitted model presented a greater coefficient of determination ($R^2$) than that of the original GFMC model. Nevertheless, both models showed a tendency to underestimate values in the warmest hours of the day. A time series comparing the original GFMC model and the fitted GFMC model for the two sampling periods is plotted in Figure 4.

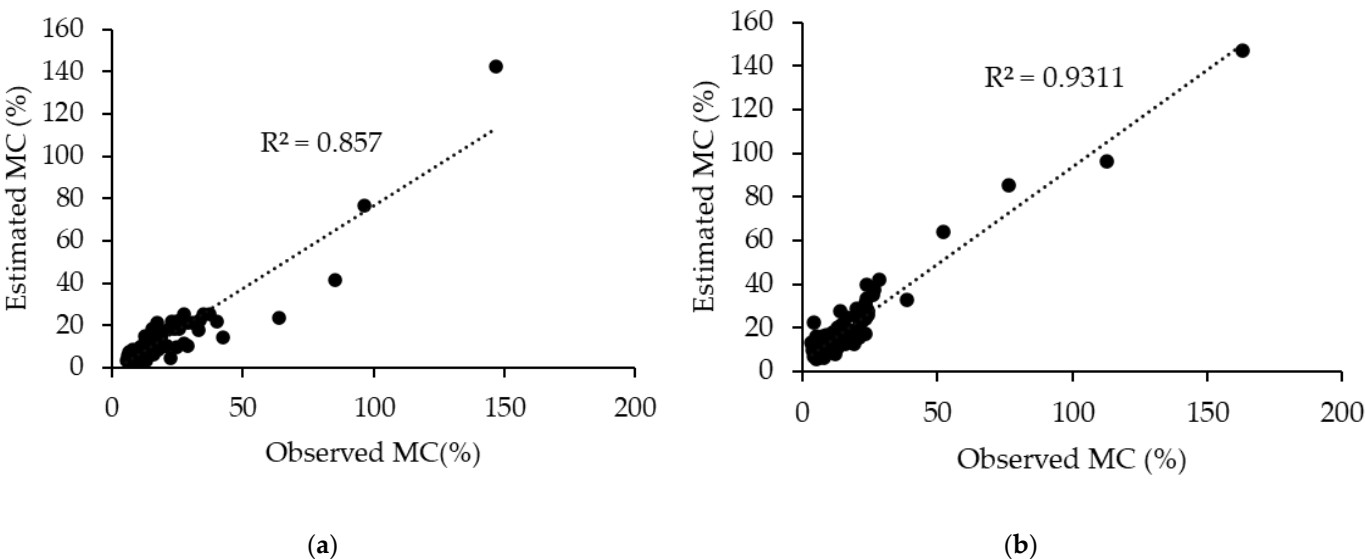

**Figure 3.** Predicted versus observed Vila Velha State Park grassland moisture content for (**a**) the original GFMC (**b**) and the fitted GFMC models.

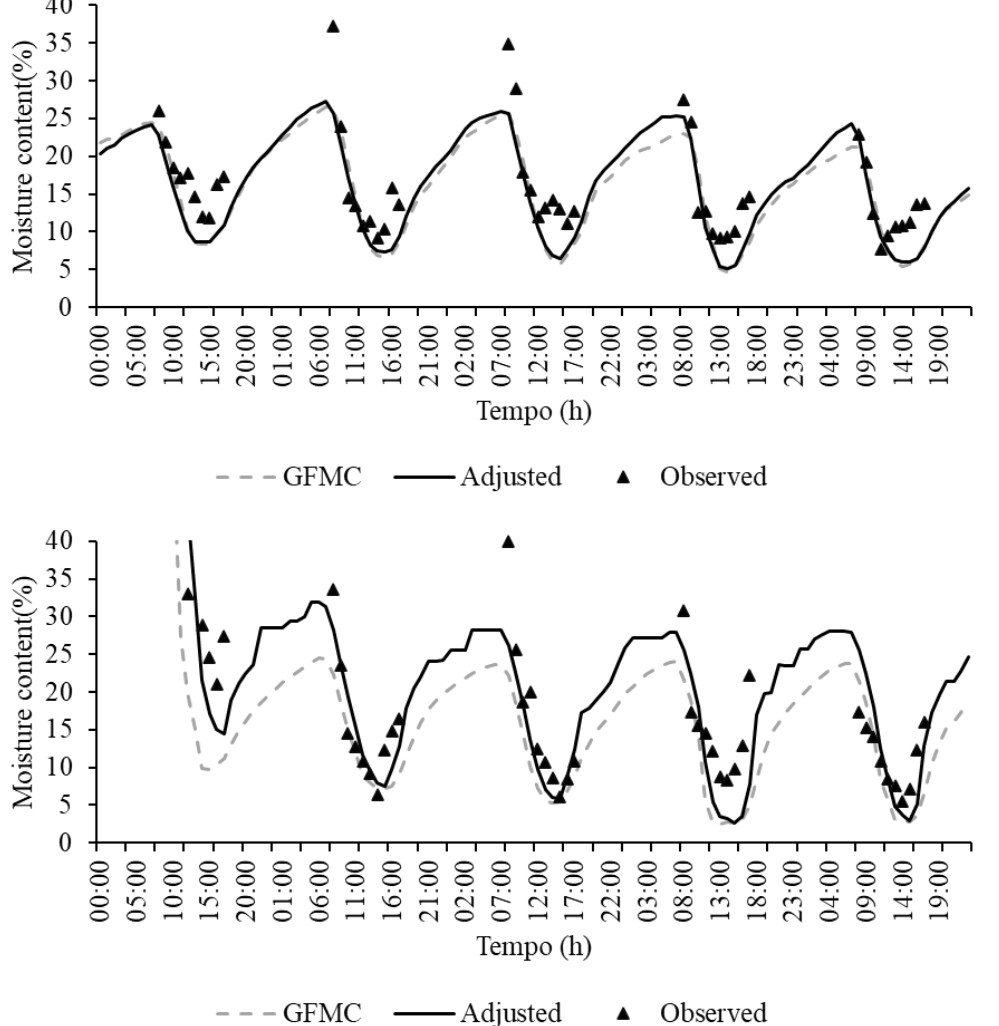

**Figure 4.** Time series from the two sampling periods of observed Vila Velha State Park grassland moisture content and predicted moisture content from the original GFMC and the fitted GFMC models.

The two models are significantly correlated with field data and show a good fit to the observations. However, the fitted GFMC model was better than the original GFMC model in all the evaluation parameters, as seen in Table 4.

**Table 4.** Summary of linear regression analysis of the original GFMC and the fitted GFMC models.

| Model | Evaluation Paramenters | | | |
|---|---|---|---|---|
| | MAE | RMSE | MBE | $R^2$ |
| GFMC | 4.3682 | 6.6881 | 0.0000 | 0.8570 |
| Fitted | 3.4272 | 4.5624 | 0.0000 | 0.9311 |

## 4. Discussion

The construction of a climatic chamber, based on recommendations in the study by Lopes [7], enabled the experiments to be performed with a fine control of the environmental conditions in order to obtain the equilibrium moisture content of Vila Velha State Park's grass fuel. The use of two digital scales made it possible to process more fuel samples simultaneously.

The exponential behavior of the sorption curve was observed in this study. However, the curves obtained for the grass fuel analyzed do not present pure exponential behaviors, as observed by Miller [15] and Lopes [6].

The adsorption process, until the fuel reaches the equilibrium moisture, is generally greater than that of the desorption process [6,16,17]. The results obtained in this work reinforce this premise, because the grass fuel took 14.93 h to lose moisture until 1.8% of the water fraction remained in the desorption process, and 15.99 h to adsorb 98.2% of its mass to the amount of water vapor available in the chamber. Anderson [16] found values of 9.24 h for desorption and 7.07 h for adsorption for *Bromus tectorum* L. and 18.32 h for desorption and 14.53 h for adsorption for *Pseudoroegneria spicata* (Pursh) Á. Love. In recent studies, Lopes [6] found values of 17.73 h for adsorption in *P. pinaster* and Bakšić [9] found 27.14 h for *P. halepensis* needles to lose 98.2% of their moisture, while Zhang and Tian [18], found values ranging from 10.42–27.45 h, studying the time lag for forests mostly composed of *Pinus yunnanensis*. These variations in response time can be explained by the presence of waxes or resins, as well the anatomical characteristics and meteorological conditions to which fine forest fuels are under [16,17].

It is observed that the first three time lags have an increasing pattern ($\tau_1 < \tau_2 < \tau_3$), constituting similar behaviors to that found by Anderson [19], Lopes [6] and Bakšić [9]. The fourth response time had the highest standard deviation for both processes. This occurs due to a high fluctuation when the fuel reaches moisture content values near to the equilibrium moisture, as the atmospheric conditions are not static [6]. In addition to the moisture exchange processes being a combination of the exponential curves, Simard [20] added that there is a sinusoidal variation within the sorption processes.

As verified, the two sorption models presented a reasonable fit to the data obtained in the laboratory for the adsorption curves. It can be assumed that both models can be used in weather conditions where the moisture content changes in a temporal base less than $\tau_1$. Nevertheless, the model developed by Henderson and Pabis was chosen for presenting the lowest RSME value.

As expected, the equilibrium moisture content data showed sigmoidal behavior. The desorption curve showed higher values than the adsorption curve, characterizing the typical hysteresis present in fine fuels. Studies about the equilibrium moisture content of fine fuels, such as Van Wagner [8], Lopes [6], Bakšić [9] and Miller [15] showed similar results to those obtained in this work. The analysis showed that the statistical evaluation of the *EMC* models corroborates the values found by Bakšić [9] for the same models.

The model adjusted in estimates, starting from 40% relative humidity, provides higher equilibrium moisture values than the original Van Wagner model. This result shows that

an adjustment for each fuel is necessary when the model is applied in places with different conditions and different types of fuel, or it can produce erroneous results [6].

The tendency to underestimate values in the warmest hours of the day is possibly linked to errors in the equilibrium moisture content estimates or due to errors in the models for determining the fuel temperature [1]. As mentioned, determining the fuel temperature in modelling the radiation influence over temperature to the reality of the southern Brazil grassland perhaps solves the problem of underestimation. The statistical parameters showed that the MAE and RMSE values obtained for the fitted GFMC model were greater than those found by Bakšić [9]. Nevertheless, it is important to point out that dew precipitation occurred during the sampling period, and this variable is not considered by the GFMC model structure and the weather system. The fitted GFMC model is unbiased due to showing values of zero for a mean bias error. The fitted GFMC model obtained higher correlations with the data collected than those found by Wotton [1] for the Echo Bay region in Canada. The fitted GFMC model also showed similar values when compared with better values of $R^2$ if compared with other studies, such as by Cruz [2] and Cawson [21]. The use of the improved model for the moisture content estimation for specific regions allows for the improvement in the abilities of combat agencies to prevent and to more accurately predict and plan the fire season; providing a safe conduction of controlled burns, an accurate prediction of fire behavior, with a provision of information to the community [22].

## 5. Conclusions

This work reinforces the robustness of the predictions obtained by models based on the Hourly Fine Moisture Code models, mainly if they are specie-specific adjusted. The estimates of the chosen equilibrium moisture and response time models demonstrated good fitting to the data obtained in the field samples and the experiments carried out in the laboratory. Despite the presence of an underestimation trend in the moisture content estimates, the model exhibited a good performance according to the statistical parameters evaluated. The fitted GFMC model is suitable to predict the fuel moisture content in highland grasslands in southern Brazil. It is correct to affirm that the hypothesis tested in this work is true and the results obtained will serve to develop a forest fire danger rating system in Brazil in the future.

**Author Contributions:** Data curation, J.F.L.d.S., B.K. and T.d.S.F.; Formal analysis, J.F.L.d.S.; Methodology J.F.L.d.S.; Writing—original draft, J.F.L.d.S., B.K. and T.d.S.F.; Supervision—review and editing, A.C.B. and A.F.T. All authors have read and agreed to the published version of the manuscript.

**Funding:** This study was partly financed by the Coordenação de Aperfeiçoamento de Pessoal de Nível Superior—Brasil (CAPES)—Finance Code 001.

**Institutional Review Board Statement:** Not applicable.

**Informed Consent Statement:** Not applicable.

**Data Availability Statement:** Not applicable.

**Conflicts of Interest:** The authors declare no conflict of interest.

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
