# Peer review of "Adjustment of the Grass Fuel Moisture Code for Grasslands in Southern Brazil"

_fire, doi:10.3390/fire5060209_

Round 1
Reviewer 1 Report
This study examines the use of two different versions of grass fuel moisture code models to moisture content of grasses across the state park of Vila Velha, Paraná, Brazil. This work will likely be important to predictions of fire danger throughout this region and vegetation type, and potentially other grassland systems as well. However, the authors do not clearly elucidate their hypothesis or question(s) for this work, and could also spend considerably more time contextualizing the importance of their findings and their relevance to scientific or applied questions in moisture modeling or fire danger prediction.
Line 18 ” perform sorption tests on climatic chamber” seems grammatically odd
Lines 22-23 “The results showed that it is recommended “ This seems awkward – I would combine the last two sentences
Line 27 – it’s odd to give precedence to grass ‘being several places on the planet’ as nothing about that intrinsically implies that its dangerous or flammable.
Line 66 – is there a reason to believe that this system is different or requires different modeling techniques? What is the question being addressed?
Fig. 4. Altering the x-axis to make the time of day more easily readable might aid with this figure. The time of day on which it began is critical, etc., – particularly as given the need to address the underestimation of moisture during the warmest hours of the day (lines 284-286.).
Lines 293-296: What are the implications of the fact that the GFMC model performed better? Some explicit explanation of why this finding is important, and what scientific or applied ramifications this finding has, would be helpful here. It’s mentioned earlier that this is a high-danger fuel – does this finding have implications for fire risk or management?
Line 306 – This may be worth expanding on in the discussion. What would the repercussions be of failing to incorporate these methods into such a danger rating system?
Author Response
Point 1 : Line 18 ” perform sorption tests on climatic chamber” seems grammatically odd.
Response 1: Sentence corrected in the text of the paper.
Point 2 : Line 27 – it’s odd to give precedence to grass ‘being several places on the planet’ as nothing about that intrinsically implies that its dangerous or flammable.
Response 2: Sentence corrected in the text of the paper.
Point 3 : Line 27 – it’s odd to give precedence to grass ‘being several places on the planet’ as nothing about that intrinsically implies that its dangerous or flammable.
Response 3: One of the reasons why we chose to adjust the GFMC for grasslands in southern Brazil (fire-dependent ecosystem) was the fact that the GFMC is a well-known model around the world. Another reason, and the main one, is the Brazil's delay in moisture content modeling research and the need to replicate this knowledge in our context. The hypothesis was changed in the text of paper.
Point 4: Fig. 4. Altering the x-axis to make the time of day more easily readable might aid with this figure. The time of day on which it began is critical, etc., – particularly as given the need to address the underestimation of moisture during the warmest hours of the day (lines 284-286.).
Response 4: The x-axis was altered.
Point 5: Lines 293-296: What are the implications of the fact that the GFMC model performed better? Some explicit explanation of why this finding is important, and what scientific or applied ramifications this finding has, would be helpful here. It’s mentioned earlier that this is a high-danger fuel – does this finding have implications for fire risk or management?
Response 5: In the last paragraph of the discussion, the main implications that of an improved model can bring were placed.
Point 6: Line 306 – This may be worth expanding on in the discussion. What would the repercussions be of failing to incorporate these methods into such a danger rating system?
Response 6: We believe that this discussion does not fit within the scope of this specific work. However, we are developing a derivative work that measures the impact of using this new system.
Reviewer 2 Report
The study is well done and clear in the presentation of the methodologies and results.
The introduction should be integrated with another bibliography. in the state of Paranà other studies have been conducted and the subject of studies on Fuel Moisture, such as doi: 10.3832 / ifor0489-002.
It is advisable to highlight the importance of knowledge of fuel models from ground research in order to validate models based on remote sensing data in order to produce useful maps for land management, bibliography recommended by citation:
doi: 10.3390 /
rs13061189
doi: 10.3832 / ifor3587-013
doi: /10.1080/01431160410001735085
Author Response
The suggestions for including the importance of field sampling and the study carried out in the Paraná region were included in the article's introduction.
Reviewer 3 Report
This manuscript provided a detailed experiment to fit the Grass Fuel Moisture Code model in southern Brazils. The estimates of the equilibrium moisture and response time models demonstrated good fitting to the data obtained in the field samples and experiments carried out in the laboratory. However, this work needs minor revision to address some critical points. They are listed below.
(1) Page 3, Equation 1 and equation 2: What is the difference between these equations in the application? Why are they used here?
(2) Page 3, Line 128: Why do you divide the sorption curves that way? On what basis?
(3) Page 3, Line 145: “Only the second stage was considered in this work”, how should your model deal with precipitation?
(4) Page 4, Equation 5: Please check if there is a “T” missing after ??-0.036. Or your wetting process is temperature-independent?
(5) Page 3, Line 136 and Page 4, Line 157: The sentence “? is temperature (°C), ? is RH” is repeated.
(6) Page 7, Figure 3, and Page 9, Line 290: You mentioned that dew precipitation occurred during the sampling period. Were those high MC values (>50%) all affected by dew precipitation? If you remove these dew-affected samples, can your fitted GFMC model perform better than the original GFMC model?
Author Response
Point 1: Page 3, Equation 1 and equation 2: What is the difference between these equations in the application? Why are they used here?
Response 1: The Byram's pure exponential equation is the first model to describe the fuel drying and wetting process. The Henderson and Pabis model works in the same way, but with the addition of a coefficient for a better fit of the model to the data obtained in the tests. Like Lopes et al (2014), we sought the best fit. However, we chose to test only the two due to the ease of adjustment and implementation in the lograte drying and wetting equations.
Point 2: Page 3, Line 128: Why do you divide the sorption curves that way? On what basis?
Response 2: This approach is adopted by the articles in the bibliography that address the study of timelag. this decision facilitated the comparison of results and standardized the study. Another reason is the fact that only the first timelag happens in real meteorological conditions as described by Lopes et al (2014).
Point 3: Page 3, Line 145: “Only the second stage was considered in this work”, how should your model deal with precipitation?
Response 3: We deal with precipitation in the same way as Wotton (2009). Due to this, the modeling only applies to the drying phase of the GFMC. Is the same procedure adopted by Bakšić (2017).
Point 4: Page 4, Equation 5: Please check if there is a “T” missing after ??-0.036. Or your wetting process is temperature-independent?
Response 4: Equation (5) was checked and the variable temperature (T) is present in both equations.
Point 5: Page 3, Line 136 and Page 4, Line 157: The sentence “? is temperature (°C), ? is RH” is repeated.
Response 5: The sentence in line 157 was removed from the article.
Point 6: Page 7, Figure 3, and Page 9, Line 290: You mentioned that dew precipitation occurred during the sampling period. Were those high MC values (>50%) all affected by dew precipitation? If you remove these dew-affected samples, can your fitted GFMC model perform better than the original GFMC model?
Response 6: A quick answer would be: probably yes! However, we chose to keep the samples influenced by dew to show a better portrait of the meteorological behavior of the region. In future works we intend to include the improvement of the estimation of dew precipitation to the model.
Reviewer 4 Report
This manuscript uses an equilibrium moisture content model to fit a grassland fuel moisture code model to estimate the moisture content for the grassland of the State Park of Vila Velha, Paraná, Brazil. This study included field sampling and laboratory processing, and through careful computational derivation and modeling, the newly fitted model achieved an improvement in accuracy over the original model. I recommend major revisions prior to publication. Some comments that might improve this work are listed below.
1) Page 4, line 151, what do kd and kw represent and how are their equations derived?
2) Page 4, line 161, Why do the drying/wetting rate equations consider only the first time lag of each process?
3) Page 6, figure 2, is the horizontal coordinate "relative humidity"? Please confirm the correct English spelling.
4) Page 4, line 155, page 6, line 215, please make sure that these two equations are correct.
5) Page 7, figure 3, please make sure that the horizontal scale is displayed completely.
6) How is the GFMC derived from the EMC model? It would be good if the relationship between GFMC and EMC could be described in detail in the article.
7) Is it possible to elaborate on the construction details of the original GFMC model and the fitted GFMC model and what are the specific improvements of the fitted model from the original model?
8) The study area of this experiment is limited to a single site, so can this research be applied to samples from other sites or other vegetation types?
9) How many samples were used in the experiment and how many samples were used for model validation? What are the meteorological conditions corresponding to the samples?
10) Page 1, line 19, What are the effects of different time lags on the model and how do they contribute to the estimation of FFMC?
11) Several references below may be considered in this work.
Cruz, M. G., S. Kidnie, S. Matthews, R. J. Hurley, A. Slijepcevic, D. Nichols, and J. S. Gould, 2016: Evaluation of the predictive capacity of dead fuel moisture models for Eastern Australia grasslands. International Journal of Wildland Fire, 25, 995.
Nolan, R. H., V. Resco de Dios, M. M. Boer, G. Caccamo, M. L. Goulden, and R. A. Bradstock, 2016: Predicting dead fine fuel moisture at regional scales using vapour pressure deficit from MODIS and gridded weather data. Remote Sensing of Environment, 174, 100-108.
Quan, X., M. Yebra, D. Riaño, B. He, G. Lai, and X. Liu, 2021: Global fuel moisture content mapping from MODIS. Int J Appl Earth Obs, 101.
Slijepcevic, A., W. Anderson, S. Matthews, and D. Anderson, 2015: Evaluating models to predict daily fine fuel moisture content in eucalypt forest. Forest Ecology and Management, 335, 261-269.
Round 2
Reviewer 4 Report
The authors addressed my questions and I recommend the publication of this work.